# Tumor-Derived Extracellular Vesicles as Liquid Biopsy for Diagnosis and Prognosis of Solid Tumors: Their Clinical Utility and Reliability as Tumor Biomarkers

**DOI:** 10.3390/cancers16132462

**Published:** 2024-07-05

**Authors:** Prerna Dabral, Nobel Bhasin, Manish Ranjan, Maysoon M. Makhlouf, Zakaria Y. Abd Elmageed

**Affiliations:** 1Vitalant Research Institute, University of California San Francisco, San Francisco, CA 94105, USA; pdabral@vitalant.org; 2Human Genome Sequencing Center, Baylor College of Medicine, Houston, TX 77030, USA; nobel.bhasin@bcm.edu; 3Department of Pediatrics, Baylor College of Medicine, Houston, TX 77030, USA; manish.ranjan@bcm.edu; 4Department of Biomedical Sciences, Discipline of Pharmacology, Edward Via College of Osteopathic Medicine (VCOM), 4408 Bon Aire Drive, Monroe, LA 71203, USA; mm.arc.sci@gmail.com

**Keywords:** tumor biomarkers, liquid biopsy, extracellular vesicles, isolation, cargo molecules

## Abstract

**Simple Summary:**

The ongoing research of extracellular vehicles (EVs including exosomes, ectosomes, and apoptotic bodies) is gaining momentum to understand these vesicles’ biology and clinical applications in cancer disease. The current limitations of using standard tumor biomarkers warrant the development of novel and reliable biomarkers to meet clinical needs. Exosomes are used as tumor biomarkers, for targeted therapy, for vaccine development, and as a vehicle for drug delivery. Here, we summarized the current approaches for different methods of EV isolation and EV cargo compositions, such as nucleic acids, proteins, and lipids. The unique cargo composition of exosomes makes it a potential candidate for liquid biopsies in the diagnosis and prognosis of cancer patients. Furthermore, the review highlights the use of machine learning algorithms to analyze complex EV datasets and create more robust models for biomarker discovery.

**Abstract:**

Early cancer detection and accurate monitoring are crucial to ensure increased patient survival. Recent research has focused on developing non-invasive biomarkers to diagnose cancer early and monitor disease progression at low cost and risk. Extracellular vesicles (EVs), nanosized particles secreted into extracellular spaces by most cell types, are gaining immense popularity as novel biomarker candidates for liquid cancer biopsy, as they can transport bioactive cargo to distant sites and facilitate intercellular communications. A literature search was conducted to discuss the current approaches for EV isolation and the advances in using EV-associated proteins, miRNA, mRNA, DNA, and lipids as liquid biopsies. We discussed the advantages and challenges of using these vesicles in clinical applications. Moreover, recent advancements in machine learning as a novel tool for tumor marker discovery are also highlighted.

## 1. Introduction

Extracellular vesicles are nanosized, lipid-bound membrane-derived vesicles released by almost all the cells into extracellular space under physiological and pathological conditions [1]. After release, they circulate in body fluids, including blood, plasma, saliva, breast milk, urine, and cerebral spinal fluid [2]. Based upon their cellular origin, function, size, and content, EVs can be classified into three major subtypes: (1) small extracellular vesicles (sEVs) or exosomes, which range from 30–150 nm in diameter. They are derived from the inward budding of the endosomal membrane; (2) microvesicles (MVs), also referred to as ectosomes, shedding vesicles or microparticles, range from 100–1000 nm in diameter. MVs originate from the budding of the plasma membrane, and (3) apoptotic bodies range from 1000–5000 nm in diameter. Apoptotic bodies arise from the plasma membrane during programmed cell death [3,4,5]. The biosynthesis of sEVs begins from the invagination of the endosomal membrane to form multivesicular bodies. Lysosomes degrade these bodies or can be released outside the cell as sEVs [6]. Ectosomes originate from the outward budding of the plasma membrane, whereas apoptotic bodies originate from apoptotic cells. The origins of exosomes and ectosomes are different, and although their contents have some similarities, each type has its own unique membrane and cargo contents (reviewed in [7,8]). EVs can carry highly heterogeneous content, including lipids, proteins, DNA, mRNA, and non-coding RNA, including microRNAs (miRNAs or miRs), to adjacent or distant cells while retaining markers specific to their cell of origin. The pathophysiological roles of EVs have gained considerable attention due to their ability to facilitate cell-to-cell communication, transport of bioactive cargo, cellular homeostasis, inflammation, and their abundance in the circulating biofluids [9]. EVs have gained extensive popularity due to their ability to regulate various aspects of cancer progression, such as cancer cell proliferation, chemoresistance, metastasis, angiogenesis, and immune system modulation. Tumor cell-associated EVs are usually distinct from those derived from normal cells concerning their number, morphology, functions, and bioactive content (proteins, miRNAs, and DNA), making them ideal candidates to develop biomarkers for cancer diagnosis and progression [10]. 

Early detection is critical for reducing patient death associated with cancer, the second leading cause of mortality worldwide. The current method of diagnosis and monitoring cancer treatment response involves biopsy, which is highly invasive, might not accurately represent tumors in the heterogenous tissue, and can fail to diagnose metastasis at secondary sites, making it hard to detect cancer at the advanced stages. Moreover, it has been suggested that this procedure could expose the patients to the risk of developing metastasis and enhanced tumor growth.

The use of biofluids such as blood or urine to detect cancer-derived molecules for the detection and screening of cancer, as well as monitoring of cancer progression, is referred to as liquid biopsy. Liquid biopsies allow for easy, non-invasive, and frequent sample collection, which can help monitor cancer progression and its response to chemotherapeutic agents, contributing to a faster evaluation and design of cancer treatment [11]. Currently, circulating tumor DNA (ctDNA) and circulating tumor cells (CTCs) are the only analytes approved by the US Food and Drug Administration (FDA) for the diagnosis and screening of cancer [12]. 

Due to the ubiquitous presence and relative abundance of cancer-derived circulating EVs in various biofluids, they are being widely studied as novel analytes for liquid biopsies. Moreover, the ability of EVs to carry bio-materials (DNA, miRNA, and proteins) on their surface and lumen, some of which are fingerprints of their cell of origin, makes EV-based liquid biopsy highly advantageous. Since exosomes remain the most studied component of EVs, in this review article, we summarized the current knowledge about commonly used methods for the isolation of EVs and discussed the recent advances in the use of EVs as circulating biomarkers for the diagnosis and prognosis of cancer. 

## 2. Methods of EV Isolations

The success of EVs as liquid biopsies is highly dependent on the choice of EV isolation method, as shown in Table 1 and Figure 1. While pure preparation is critical to ensure the success of downstream applications, other factors are to be considered when deciding on a method of isolation, including yield and integrity of the purified EVs and the processing time for subsequent analysis. Moreover, specimen handling conditions must also be optimized, including source and collection of samples, storage conditions, and preparation. EV liquid biopsy specimens include blood serum or plasma, urine, cerebrospinal fluid, peritoneal fluid, pleural effusion, and tears. Present isolation procedures suffer limitations due to ambiguous definitions and nomenclature of EV subtypes, loss of yield and purity, and damage to EV structures [13]. These methods used different physical and biological properties, including size, shape, density, charge, and antigen exposure.

Ultracentrifugation is one of the most commonly used procedures to isolate EVs, and it can be divided into differential centrifugation and density gradient centrifugation. Differential ultracentrifugation, a gold standard for EV isolation, uses centrifugal force to pellet EVs based on size and density. Density gradient centrifugation isolates EVs based on their size/density or both using a density gradient commonly generated by sucrose or iodixanol. These ultracentrifugation techniques are time-consuming, laborious, and do not yield pure preparations. Ultrafiltration and Size Exclusion Chromatography purifies EVs based on their particle size. While ultrafiltration can be used to concentrate EVs from a large volume and is usually time efficient, it could lead to sample loss and potential contamination of proteins. Size exclusion chromatography enables efficient separation of EVs from small sample volumes based on size and preserves their structural integrity and bioactivity [14]. However, it may require prior purification steps and a longer processing time. Polymer-based precipitation is another method to isolate EVs using water-excluding polymers like polyethylene glycol (PEG), which reduce the solubility of EVs and lead to their settling out of solution through low-speed centrifugation. It is an attractive choice for EV isolation as it is fast, requires no specialized equipment, and can be used with large sample volumes. Though this procedure results in a high yield of EVs, it can lead to co-precipitation of other contaminants, resulting in an impure preparation (reviewed in [15]). Immunoaffinity capture-based techniques exploit the interaction between EV membrane proteins and antibodies immobilized on beads or matrices, resulting in a highly specific, pure, rapid isolation of desired EVs: high cost, damage to EV structures, elution from beads, and low yield. An innovative, rapid, and efficient technique for the recovery of EVs is a microfluidics-based approach. Microfluids manipulate small volumes of liquids in microsized channels using distinctive physical and biochemical properties like size, density, and immune interactions. Extensive research must be carried out to overcome limitations such as high cost, additional equipment, trained personnel, and damage to the EV structures while recovering [16,17].

**Table 1 cancers-16-02462-t001:** Methods of Extracellular Vesicles (EVs) isolation from different body fluids.

Types	Ultracentrifugation		Ultrafiltration	Size Exclusion Chromatography	Polymer-Based Precipitation	Immunoaffinity Capture	Microfluidics-Based Approach
	Differential Centrifugation	Density Gradient Centrifugation					
Description	Centrifugal force to pellet EVs	Density gradient by sucrose or iodixanol	Isolation based on pore size or M.Wt. cut off of the membrane	Polymer-based method that allows particles of different sizes to be differentially eluted by the chromatography system	Water-excluding polymers like polyethylene glycol (PEG)	Interaction between EVs membrane proteins and antibodies immobilized on beads or matrices	Small volumes of liquids in microsized channels with specific physical and biochemical properties
Based on	Size and density	Size/density or both	Size		Solubility and size	EVs protein markers	Size, density, and immune interactions
Advantages	▪ Standard method ▪ High yield		▪ Time efficient ▪ Concentrates EVs from large sample volume	▪ Efficient separation from small sample volume ▪ Preserves EV’s structural integrity and bioactivity	▪ Fast method ▪ Reduces EVs solubility using low-speed centrifugation ▪ Does not require special equipment ▪ Can be used with large sample volumes ▪ High yield	▪ Fast method ▪ Highly specific ▪ High pure yield	▪ Fast and Efficient
Disadvantages	▪ Time-consuming ▪ Laborious ▪ Do not yield pure EVs		▪ Loss of sample ▪ Potential contamination of other proteins	▪ Requires prior steps of purification ▪ Longer processing time	▪ May co-precipitate other contaminants	▪ High cost ▪ Can damage EV structure ▪ Elution from beads ▪ Low yield	▪ High cost ▪ Needs additional equipment ▪ Needs trained personnel ▪ Can damage EV structure
References	[14]				[15]	[16,17]	

## 3. Use of EVs as Liquid Biopsy

sEVs and circulating tumor DNA (ctDNA) are both essential components of liquid biopsies used in cancer diagnostics and prognostications [18,19]. However, sEVs offer several advantages over ctDNA. The sEV cargo contains RNA that increases the number of mutant copies available for sampling compared to ctDNA alone [20]. The homogeneous size of these vesicles makes their detection easy by electron microscopy [21]. The cargo contents of lipid bilayer sEVs are more stable, making them more robust for analysis than ctDNA [22]. Moreover, the possibility of identifying gene mutations in sEVs is higher than in ctDNA [23]. Additionally, sEV-associated mRNA is actively released from donor cells compared to ctDNA released by necrotic or apoptotic cells [24]. Furthermore, using sEV-associated RNA combined with either cfDNA or circulating tumor cells (CTCs) in liquid biopsies has shown promise in identifying somatic mutations of tumor origin [25,26]. 

### 3.1. Proteins

sEVs carry multiple proteins, some of which are specific to the cell of their origin, whereas others are conserved across all exosomes [8]. These proteins play an important role in recognizing recipient cells for transferring bioactive content and regulating the sorting of EV components (summarized in Table 2). A recent report by Melo et al. used Mass Spectrometry to detect overexpression of Glypican-1 (GPC1), a cell surface expressing glycoprotein, in cancer-derived exosomes. Using flow cytometry, GPC1+ exosomes were highly enriched serum samples from patients suffering from Pancreatic Ductal Adenocarcinoma (PDAC). Interestingly, they also established the importance of GPC1+ circulating exosomes as a diagnostic marker in early-stage pancreatic cancer and a prognostic marker to monitor survival post-surgery [27]. GPC1+ exosomes were also reported to be ten-fold elevated in plasma samples collected from colorectal cancer patients compared to healthy controls [28]. Another 2017 report identified ephrin type-A receptor 2 (EphA2) as a candidate biomarker for pancreatic cancer. They used the EphA2 antibody to detect EphA2 positive exosomes in blood plasma using a gold nanoparticle-based nanoplasmon-enhanced scattering (nPES) assay, which used the plasmon effect to detect sEVs. Eph2A-positive exosome levels were enriched in samples collected from early-stage pancreatic cancer patients compared to normal controls or patients with pancreatitis [29]. Similarly, migration inhibitory factor (MIF) was highly enriched in plasma samples from PDAC patients compared to healthy controls or patients who have been disease-free for over 5 years. Moreover, they also reported high MIF levels in exosomes before liver metastasis, adding to the clinical value of MIF as a prognostic marker [30]. 

Leucine-rich a-2-glycoprotein (LRG1) was reported to be highly overexpressed in urinary exosomes of patients suffering from Non-Small Cell Lung Cancer (NSCLC) using mass spectrometry. These results were validated using Western blotting of exosome samples and immunohistochemistry of lung tissue, which showed higher LRG1 expression in urinary exosomes and lung tissues from NSCLC patients, respectively, compared with healthy controls [31]. CD91 was also identified as a highly expressed protein on the surface of exosomes from serum and blood plasma samples of patients with advanced NSCLC using mass spectrometry and antibody microarray [32,33]. Similarly, Galectin-3-binding protein (LG3BP) and polymeric immunoglobulin receptor (PIGR), which were selectively enriched in exosomes isolated from the serum of patients suffering from liver and biliary cancer, could be utilized for diagnosis of cancer [34].

A report by Yoshioka et al. 2014 identified the diagnostic use of CD147 in colorectal cancer, as CD147 was highly expressed in exosomes from serum samples of patients with colorectal cancer compared to healthy controls [35]. Another report identified Copine 3 or CPNE3 as elevated in exosomes isolated from the plasma of colorectal cancer patients compared to healthy control. Furthermore, CPNE3 expression increased with cancer progression, and CPNE3 enhanced the diagnostic power of carcinoembryonic antigen (CEA), a previously identified biomarker, when used in conjunction [36].

Survivin was reported to be a possible biomarker for the diagnosis and prognosis of prostate cancer, as survivin levels were elevated in exosomes isolated from plasma/serum samples of prostate cancer patients as compared to those isolated from healthy males [37]. Two studies reported the use of prostate-specific antigen (PSA) and gamma-glutamyltransferase 1 (GGT1) to distinguish between normal or benign prostatic hyperplasia and prostate cancer patients, as the PSA and GGT1 protein levels were markedly higher in exosomes isolated from the blood of prostate cancer patients [38,39]. Our previous study demonstrated that exosomal ITGA2 was highly enriched in the plasma collected from prostate cancer patients compared to non-cancerous subjects [40].

In another study, mass spectrometry and ELISA were used to establish the association of tumor-associated calcium-signal transducer 2 (TACSTD2) with bladder cancer in urinary exosomes [41]. A surface plasmon resonance-based assay to detect exosomes was utilized to report elevated levels of epithelial cell adhesion molecule (EpCAM) and CD24 in exosomes from ascites of ovarian cancer patients, indicating their use as a diagnostic and prognostic biomarker [42]. The cerebrospinal fluid of brain tumor patients also showed higher levels of Interleukin 13 Receptor alpha 2 (IL13Rα2) when quantified using flow cytometry [43]. 

**Table 2 cancers-16-02462-t002:** List of Extracellular Vesicle (EV)-associated proteins as tumor biomarkers.

Cancer Type	Protein-Based Biomarker	Isolated From	Method of Isolation	Biomarker Type	Significance	References
Bladder Cancer	Tumor-associated calcium signal transducer 2 (TACSTD2)	Urine	- Isotopic dimethylation labeling	- Diagnostic	Patients > healthy controls	[41]
Brain Cancer	Interleukin 13 receptor subunit alpha 2 (IL13Rα2)	Cerebrospinal fluid Tissue culture	- Ultracentrifugation	- Diagnostic	Patients > healthy controls	[43]
	Epidermal growth factor receptor variant III (EGFRvIII) and TGF-β1	Serum Plasma	- Ultracentrifugation - Microfluidics-based Approach	- Diagnostic - Prognostic	Patients > healthy controls Prediction of treatment response	[44,45]
Breast Cancer	Fibronectin and Developmental endothelial locus-1 (Del-1)	Plasma	- Immunoaffinity capture using ELISA	- Diagnostic	Patients > healthy controls or patients with post-surgery resection	[46,47]
Cholangiocarcinoma	Galectin-3-binding protein (LG3BP) and Polymeric Immunoglobulin receptor (PIGR)	Serum	- Ultracentrifugation	- Diagnostic	Patients > controls	[34]
Colorectal Cancer	Glypican-1 (GPC1)	Plasma	- Ultracentrifugation	- Diagnostic	Patients > healthy controls	[28]
	CD147 (Basigin)	Serum	- Immunoaffinity capture	- Diagnostic	Patients > healthy controls	[35]
	Copine 3 (CPNE3)	Plasma	- Ultracentrifugation	- Diagnostic - Prognostic	Patients > healthy controls CRC patients with lower exosomal CPNE3 levels have better disease-free survival and overall survival	[36]
Lung Cancer	Leucine-rich alpha-2 (LRG1)	Urine	- Ultracentrifugation	- Diagnostic	Patients > healthy controls	[31]
	CD91 (LRP1)	Plasma	- Immunoaffinity capture	- Diagnostic	Patients > healthy controls	[32,33]
Melanoma	Tyrosinase-related protein-2 (TYRP2), Very late antigen 4 (VLA-4), Heat shock protein 70 (HSP70), HSP90, and Proto-oncogene c-Met (MET)	Plasma	- Ultracentrifugation	- Prognostic	Patients > healthy controls	[48]
	S100 calcium-binding protein B (S100B) and Melanoma inhibitory activity (MIA)	Serum	- Ultracentrifugation - Polymer-based Precipitation	- Diagnostic - Prognostic	Patients > healthy controls	[49]
	CD63 and Caveolin	Plasma	- Immunoaffinity capture using ELISA (Exotest)	- Diagnostic	Patients > healthy controls	[50]
Ovarian Cancer	Epithelial cell adhesion molecule (EpCAM) and CD24	Ascites Tissue culture	- Ultracentrifugation	- Diagnostic	Patients > healthy controls	[42]
Pancreatic Cancer	Glypican-1 (GPC1)	Serum	- Flow Cryometry	- Diagnostic	Patients > healthy controls	[27]
	Ephrin type-A receptor 2 (EphA2)	Plasma	- Immunoaffinity capture	- Diagnostic	Patients > healthy control and pancreatitis	[29]
	Migration inhibitory factor (MIF)	Plasma	- Ultracentrifugation	- Prognostic	Patients > healthy controls	[30]
Prostate Cancer	Survivin (IAP4)	Serum Plasma	- Ultracentrifugation	- Diagnostic - Prognostic	Patients > healthy controls Relapsed patients > controls	[37]
	Prostate-specific antigen (PSA) and Gamma-Glutamyltransferase 1 (GGT1)	Serum Tissue culture	- Ultracentrifugation - Immunoaffinity capture	- Diagnostic	Patients with prostate cancer > benign prostatic hyperplasia	[38,39]
	Integrin subunit alpha 2 (ITGA2)	Plasma	- Ultracentrifugation - Polymer-based Precipitation	- Diagnostic	Patients > healthy controls	[40]
Renal cell carcinoma	Carbonic anhydrase IX (CAIX), Matrix metalloproteinase 9 (MMP-9), Dickkopf related protein 4 (DKK4), Ceruloplasmin (CP), Podocalyxin (PODXL), and Extracellular matrix metalloproteinase inducer (EMMPRIN)	Urine	- Ultracentrifugation	- Diagnostic	Patients > healthy controls	[51]

Human blood plasma samples from melanoma patients had elevated levels of tyrosinase-related protein-2 (TYRP2), very late antigen 4 (VLA-4), heat shock protein 70 (HSP70), an HSP90 isoform, and MET oncoprotein in the exosomes detected using a combination of electron microscopy and Western blotting. Moreover, it was reported that co-expression of MET and TYRP2 in exosomes could be used as a prognostic marker, and high levels of MET and TYRP2 were observed during melanoma progression [48]. Similarly, S100 calcium-binding protein B (S100B) and Melanoma Inhibitory Activity (MIA) protein levels were highly elevated in serum exosomes of advanced-stage melanoma patients compared to healthy and disease-free controls when detected by ELISA using specific antibodies [49]. CD63 and caveolin were also identified to be enriched in exosomes from the plasma of melanoma patients, using a combination of ELISA, Western blotting, and flowcytometry [50]. 

Mass spectrometry and Western blotting were used to identify the exosomal proteins in urine samples of renal cell carcinoma patients, and these could be distinguished from healthy controls due to their differential expression. These include Carbonic Anhydrase IX (CAIX), Matrix metalloproteinase 9 (MMP-9), Dickkopf-related protein 4 (DKK4), Ceruloplasmin (CP) and Podocalyxin (PODXL) which were relatively abundant in urinary exosomes of RCC patients and Extracellular Matrix Metalloproteinase Inducer (EMMPRIN) which was significantly reduced in RCC patients [51]. EGFRvIII, genomic variant III of epidermal growth factor receptor (EGFR), was also identified as an effective diagnostic and prognostic biomarker for glioblastoma. This protein was highly upregulated in sEVs isolated from sera and plasma samples of glioblastoma patients using Western blotting and nuclear magnetic resonance systems [44,45]. Fibronectin was reported to be elevated in EVs isolated from the blood plasma of advanced-stage breast cancer patients. In contrast, developmental endothelial locus-1 protein (Del-1) was abundant in EVs isolated from plasma samples of early-stage breast cancer patients compared to healthy controls or post-surgery patients [46,47]. Though protein-based biomarkers have been extensively popular among EV biomarkers, their development faces challenges when working with complex samples like plasma or serum, where a high abundance of non-vesicular proteins makes the isolation of low abundance protein complex and the presence of heterogenous posttranslational modification, which adds to the complexity of the sample.

### 3.2. miRNAs

The method of RNA isolation may affect the sEVs yield, purity, and stability, especially RNA content. So, it is essential to select the method of RNA isolation according to the study design and availability of body fluids. For example, a pure column method produces a high RNA yield compared to the phenol extraction method [52]. Another research group reported that EVs-associated RNA yield is high when isolated by membrane affinity column versus conventional ultracentrifugation method [53].

miRNAs are the class of small non-coding RNAs that play a pivotal role in gene expression at the post-transcriptional level. miRNAs contribute to various biological processes, including normal and pathophysiological conditions [54]. A list of identified EV-associated miRs is summarized in Table 3. A research team reported using miR-21 as a biomarker for glioblastoma. miR-21 levels were significantly elevated in EVs isolated from the cerebrospinal fluid of glioblastoma patients compared to non-oncological patients. Moreover, miR-21 levels were found to be reduced post-surgical resection [55]. Another study reported using miR-320 and miR-574-3p, along with a small nuclear RNA, RNU6-1, as a diagnostic marker as they were elevated in the serum of glioblastoma patients compared to healthy controls [56]. Early-stage colorectal cancer patients displayed higher levels of miR-125a-3p in their exosomes isolated from plasma than healthy controls [57]. Similarly, miR-19a and miR-92a were upregulated in exosomes isolated from plasma samples of colorectal cancer patients as compared to healthy controls [58]. Some other miRNAs identified to be explicitly upregulated in the exosomes isolated from colorectal cancer patients include let-7a, miR-1224-5p, miR-1229, miR-1246, miR-150, miR-21, miR-223, and miR-23a [59].

Selected miR-1246, miR-4644, miR-3976, and miR-4306 were reported to be highly elevated in exosomes isolated from the serum of pancreatic cancer patients in comparison with healthy controls when quantified using quantitative Real-Time PCR (qPCR) [60]. Another study identified miR-17-5p and miR-21 as highly specific and sensitive biomarkers as their levels were significantly elevated in serum exosomes of pancreatic cancer patients compared to healthy individuals or chronic pancreatitis and benign pancreatic tumor patients. Moreover, the predictive value of using miR-17-5p as a biomarker was reported as its levels increased with the advanced stage of pancreatic cancer, which could help monitor disease progression and metastasis [61]. A surface plasmon resonance-based assay and qPCR were used to detect the abundance of miR-10b in plasma exosomes of pancreatic cancer patients. Interestingly, this study reports higher miR-10b levels in plasma exosomes in chronic pancreatitis patients than normal controls, whereas the highest miR-10b levels were seen in pancreatic cancer patients. Thus, miR-10b could prove to be a valuable biomarker for early diagnosis of PDAC [62]. Another report identified miR-10b, miR-21, miR-30c, miR-181a, and miR-let7a to be upregulated in exosomes from plasma samples of PDAC patients that can be useful in differentiating them from chronic pancreatitis patients and normal individuals [63]. A recent report discussed the significant role of EV-associated miRs in pancreatic cancer pathogenesis and their utility as future biomarkers and therapeutic agents [64].

**Table 3 cancers-16-02462-t003:** List of Extracellular Vesicle (EV)-associated miRNAs as tumor biomarkers.

Cancer Type	miRNA	Isolated From	Biomarker Type	Significance	References
Breast Cancer	miR-1246 miR-21	Plasma	- Diagnostic	Patients > healthy controls	[65,66]
	miR-21 miR-105	Plasma	- Prognostic	Patients > healthy controls Metastatic > non-metastatic patients	[67]
	miR-27a miR-155 miR-376a miR-376c	Plasma	- Prognostic	Predicts pathological complete response after neoadjuvant therapy	[68]
Colorectal Cancer	miR-125a-3p	Plasma	- Diagnostic	Patients > healthy controls	[57]
	miR-19a miR-92a let-7a miR-1224-5p miR-1229 miR-1246 miR-150 miR-21 miR-223 miR-23a	Plasma Serum	- Diagnostic - Prognostic	Patients > healthy controls Predicts tumor recurrence	[58,59]
Glioblastoma	miR-21	Cerebrospinal fluid	- Diagnostic	Patients > healthy controls	[55]
	miR-320 miR-574-3p	Serum	- Diagnostic	Patients > healthy controls	[56]
Hepatocellular Carcinoma	miR-1247-3p	Serum	- Prognostic	Predicts lung metastasis	[69]
	miR-18a miR-221 miR-222 miR-224	Serum	- Diagnostic	HCC patients > chronic hepatitis B and liver cirrhosis patients	[70]
Lung Cancer	let-7b-5p let-7e-5p miR-21-5p miR-24-3p	Plasma	- Diagnostic	Distinguish early-stage patients from healthy controls	[71]
	miR-151a-5p miR-30a-3p miR-200b-5p miR-629 miR-100 miR-154-3p	Plasma	- Diagnostic	Patients > healthy controls	[72]
Ovarian Cancer	miR-21	Peritoneal fluid	- Prognostic	Patients > healthy controls	[73]
	miR-30q-5p	Urine	- Diagnostic	Patients > healthy controls	[74]
Pancreatic Cancer	miR-1246 miR-4644 miR-3976 miR-4306	Serum	- Diagnostic	Patients > healthy controls	[60]
	miR-17-5p miR-21	Serum	- Prognostic	Patients > healthy controls or chronic pancreatitis and benign pancreatic tumors	[61]
	miR-10b	Plasma	- Diagnostic	Pancreatic cancer patients > chronic pancreatitis > healthy controls	[62]
	miR-10b miR-21 miR-30c miR-181a miR-let7a	Plasma	- Diagnostic	Chronic pancreatitis patients > healthy controls	[63]
Prostate Cancer	miR-375 miR-141	Plasma	- Diagnostic - Prognostic	Patients > healthy controls Disease staging	[75]
	miR-1290 miR-375	Plasma	- Prognostic	Associated with poor overall survival	[76]
	miR-6068 miR-1915-3p miR-6716-5p miR-3692-3p	Plasma	- Diagnostic - Prognostic	Patients > healthy controls Stratify patients according to Gleason score and race	[77]

miR-375 and miR-141 were highly upregulated in EVs isolated from the plasma of patients who have metastatic prostate cancer, thus suggesting their use as a biomarker to identify metastasis [75]. Similarly, miR-1247-3p levels were elevated in serum exosomes from hepatocellular carcinoma (HCC) patients suffering from lung metastasis, which could be useful in developing preventative and therapeutic treatment [69]. Huang et al. reported the use of miR-1290 and miR-375 in plasma exosomes as prognostic markers for castration-resistant prostate cancer (CRPC) since elevated levels of these miRNAs correlated with approximately 80% death rate and low levels of the miRNAs were associated with 10% death rate [76]. Another report identified upregulation of miR-18a, miR-221, miR-222, and miR-224 in serum exosomes of HCC patients compared to chronic hepatitis B and liver cirrhosis patients [70]. Our recent study showed that exosomal miRNAs isolated from the blood of prostate cancer patients can differentiate patients according to their Gleason score and race and predict their recurrence-free survival [77]. Jin et al. reported the use of let-7b-5p, let-7e-5p, miR-21-5p, and miR-24-3p, isolated from plasma exosomes, as primary diagnostic markers to distinguish between early-stage NSCLC patients and healthy individuals [71]. Another study identified miR-151a-5p, miR-30a-3p, miR-200b-5p, miR-629, miR-100, and miR-154-3p in the plasma exosomes as potential biomarkers for the diagnosis of lung adenocarcinoma [72].

A study reported high levels of miR-1246 and miR-21 in plasma exosomes of breast cancer patients as compared to healthy individuals, which could serve as effective indicators of early-stage breast cancer [65,66]. In addition, increased levels of miR-21 and miR-105 in the plasma exosomes of metastatic breast cancer patients as opposed to those from healthy individuals or breast cancer patients with non-metastatic disease, thus posing as candidates for an effective prognostic biomarker [67]. In another study, plasma exosomes exhibited high levels of miR-27a, miR-155, miR-376a, and miR-376c in breast cancer patients. Interestingly, the expression of these miRNAs was downregulated post-neoadjuvant therapy before surgery, similar to the levels in healthy controls [68].

Ovarian carcinoma patients showed high levels of miR-21 in exosomes isolated from their peritoneal fluid samples. These results were also confirmed in ovarian carcinoma and normal ovary specimens, where in situ hybridization showed high amounts of miR-21 expression in the former specimens [73]. Additionally, miR-30q-5p was highly concentrated in the exosomes isolated from urine samples of ovarian cancer patients over healthy control, where it was approximately 3-fold times lower [74].

### 3.3. mRNAs

The first study that reported the use of mRNA as biomarkers was by Skog et al., where Glioblastoma patients displayed elevated EGFRvIII mRNA, a mutant version of EGFR, in the microvesicles of their sera samples in approximately 28% of the patients. Importantly, EGFRvIII mRNA was not detectable in serum samples of healthy individuals, along with serum samples of patients who had undergone surgical removal of the tumor [78]. In a study by Yokoi et al., MMP1 (Matrix Metallopeptidase 1) mRNA was highly enriched in exosomes derived from ascites of ovarian cancer patients suffering from a high phenotype. This exosomal MMP1 mRNA was implicated in inducing apoptosis in mesothelial cells and peritoneal dissemination [79]. A similar report identifies high androgen-receptor splice variant 7 (AR-V7) mRNA and low mRNA transcripts of its total length variant in urine exosomes of advanced-stage prostate cancer [80]. 

Human telomerase reverse transcriptase (hTERT) mRNA was detected in serum exosomes of approximately 67% of cancer patients, whereas none was detected in the healthy controls. Moreover, high hTERT mRNA in the exosomes was associated with disease progression, which highlights its possible role as a pan-cancer biomarker [81], as shown in Table 4.

### 3.4. DNA

EVs have been shown to contain genomic DNA, mitochondrial DNA, single-stranded DNA, and transposable elements, which has sparked considerable interest in using DNA in circulating EVs as liquid biopsies [87]. As presented in Table 4, Thakur et al. reported the importance of exosomal DNA as a circulating biomarker for cancer diagnosis. They showed most DNA associated with the exosomes to be predominantly double-stranded, and this exosomal DNA was representative of the entire genome and reflected the mutations in parent tumor cells [88]. A study by Kahler et al. reported that exosomes in pancreatic cancer cell lines and patients’ serum contain genomic DNA fragments spanning all chromosomes, and this exosomal DNA contains mutations in KRAS and p53 genes [83]. A similar report utilized the detection of KRAS gene mutations in plasma exosomal DNA to distinguish PDAC patients from healthy controls. In addition, a KRAS mutation frequency of greater than 1% was associated with decreased survival probability of disease-free patients post-treatment, and KRAS mutation detection could predict PDAC with high sensitivity and specificity [84].

A study reported high numbers of EVs in the plasma of prostate cancer patients in comparison with healthy individuals, and these EVs showed the presence of genomic DNA fragments in different sub-types of EVs, including macrovesicles, apoptotic bodies, and exosomes [86]. Another group reported the use of exosome-based liquid biopsy where the exosome DNA isolated from peripheral blood and pleural effusion samples of pancreatic cancer patients strongly represented tumor DNA, and mutations in the NOTCH1 and BRCA2 DNA sequence were also identified [85]. Exosomal DNA from colorectal cancer cell lines displayed frameshift mutations in the microsatellite region of a tumor suppressor gene, the Transforming Growth Factor Beta Receptor Type 2 (TGFBR2) gene, which is similar to microsatellites of the cellular phenotype [82]. The analysis of genomic alterations in EV DNA during tumor progression is an attractive strategy that highlights the clinical value of EV DNA as potential biomarkers for cancer diagnosis and monitoring. EVs are highly enriched with tumor DNA compared to cell-free DNA, as the DNA enclosed in the EV membranes is relatively stable due to protection from DNases in the plasma. Moreover, the short half-life of EV DNA enables accurate representation of the dynamic tumor signature, which makes it a useful tool for long-term monitoring of tumor progression and its response to chemotherapy. 

### 3.5. Lipids

Lipids play critical roles in normal and cancer cells, and lipid metabolism is often aberrated in the latter, which contributes towards cancer progression and metastasis. Lipids are essential constituents of EV cargo and perform various functions, including maintenance of EV structures, EV biogenesis, membrane trafficking, and signaling. Though using lipids from exosomes as a biomarker is an attractive possibility, the molecular composition of exosome lipids under normal and pathobiological conditions remains highly unknown. A previous study identified 10 diagnostic biomarkers for prostate cancer patients using quantitative and qualitative profiling of urinary phospholipids [89]. Of particular interest is another report by Llorente et al., which uses quantitative lipidomics to identify potential diagnostic markers associated with prostate cancer using exosomes isolated from a highly metastatic prostate cancer cell line, PC-3. The lipid composition varied between parent cells and exosomes, where some glycosphingolipids like HexCer and LacCer were detected at elevated exosome levels [90]. The first report that used the lipidomics approach in patient samples was a preliminary study by Boccio et al., where exosomes isolated from urine samples of renal cell carcinoma patients were markedly different in their lipid composition compared to exosomes from healthy controls [91]. A similar study described the use of lipids in urinary exosomes as a potential biomarker, where they observed significant differences in the levels of nine lipid species between normal and prostate cancer patients. Moreover, a combination of three lipid species was able to diagnose the disease with a high sensitivity and specificity [92]. A recent study employed a size-based lipidomic approach to observe differences between the urinary exosomes of prostate cancer patients and those of healthy individuals. In this study, exosomes were first fractionated based on their size, after which they were examined by mass spectrometry, where most lipids were elevated twofold compared with healthy controls [93]. 

## 4. Machine Learning and Liquid Biopsy

Artificial intelligence (AI) has emerged as a favorable option for early detection of different malignancies. Machine learning (ML) is a subcategory of AI that uses algorithms to analyze collected data, learn from it, and develop models that assist in prediction and making decisions. The rapidly growing field of MI has shown potential improvement in diagnosing various diseases, including cancer. Shin et al. used AI to early detect six solid cancer types by analyzing surface-enhanced Raman spectroscopy profiles of exosomes [94]. In another study, an ML-based computational method was used to differentiate different cancer types using a panel of exosome-associated proteins [95]. In this study, highly abundant Ezrin (EZR), Talin-1 (TLN1), Adenylyl cyclase-associated protein 1 (CAP1), and Moesin (MSN) have been used as exosomal tumor biomarkers. Moreover, a deep learning model was used to improve the diagnostic accuracy of lung cancer using digital cytological images of respiratory specimens collected from over 200 multi-centers [96]. A multichannel nanofluidic system was developed to analyze RNA isolated from exosomes derived from pancreatic cancer samples, and an ML algorithm was used to generate predictive panels that were able to identify tumors from healthy controls [97].

DNA point accumulation for imaging in nanoscale topography (DNA-PAINT) and an ML algorithm have been utilized to automatically analyze data collected from four exosomal surface markers at the single-exosome level [98]. This model was able to detect breast and pancreatic cancers from unknown blood samples. ML technique was used to identify specific exosomal RNA signatures for the prediction of hepatocellular carcinoma [99]. A research team developed a novel ML model called ExoGRU to predict small RNA secretion probabilities from primary RNA sequences [100]. The model revealed cis and trans factors associated with small RNA secretion, including RNA-binding proteins. Kim et al. used nanoplasmonic spectra and a deep learning algorithm to identify mutated proteins in circulating exosomal cargo. The model used was able to locate different mutant forms of epidermal growth factor receptor (EGFR) in blood collected from lung cancer patients [101]. Standardized ML is a useful tool for cancer detection. ML models should consider the confounding factors in cancer samples and data collected from different centers to improve cancer prediction. 

## 5. Limitations and Challenges for EVs as Therapeutics in the Medical Field

Despite recent advancements in sEVs research, several barriers need to be addressed to utilize these vesicles in the era of personalized medicine. These barriers include, but are not limited to, technical issues, clinical obstacles, shared data, and the nature of conducted research [102,103,104,105,106,107,108]. Technical issues comprise the nature and volume of the collected fluids, stability of sEVs contents after collection, timing of sample collection, lack of standardized sEVs isolation and validation methods, variations in detection methods, turnaround time, and limited number of samples. Regarding clinical obstacles, there are inter- and intraindividual variations, patient comorbidities, genetic backgrounds, received medications, stage of the disease, the detection limit of the tumor, integrating biomarker data with other clinical outcomes, misleading diagnosis, and cost-effectiveness of these biomarkers. Shared data are critical to ensure the reproducibility and transparency of these data to test and validate sEV biomarkers. Given the new advancement in OMICs technology and machine learning approach, access to large datasets is essential. For research components, small-scale studies, funding availability, limited collaborations, availability of tissue specimens alongside clinical data, shared equipment, and models used for investigations limit the current efforts to develop new tumor biomarkers from sEVs. Therefore, further studies are needed to address these barriers for future sEV-based biomarkers.

## 6. Conclusions and Future Prospective

An ideal biomarker should be noninvasive, cost-effective, reproducible, and enable early disease diagnosis. Using EV proteins, miRNA, mRNA, DNA, and lipids found in body fluids could serve as liquid biopsies, ensuring a less invasive approach for cancer diagnosis, real-time monitoring of disease progression, and response to chemotherapeutic agents. sEV-derived tumor biomarkers are undergoing clinical trials and have not yet received approval from the Food and Drug Administration (FDA). However, they promise to develop novel tumor biomarkers for unmet clinical needs. To determine the most effective detection method for exosome proteins, mRNA, miRNA, DNA, and lipids, various studies have been conducted focusing on different aspects of sEV analysis. For example, detecting sEV-associated miRNAs has been a prominent area of research. Several studies have highlighted the advantages of utilizing sEVs for miRNA detection. In addition to the traditional methods for miRNA detection, innovative techniques have been developed to enhance the detection sensitivity and efficiency of sEV-associated miRNAs [109]. These methods offer advantages such as cost-effectiveness, non-invasiveness, and high sensitivity, making them promising tools for biomarker development. The developed methods continuously evolve to establish more efficient and reliable techniques for EV cargo detection.

Before using EV-based biomarkers, it is important to address some of the current challenges of using sEVs in clinical applications. Nonetheless, the potential of EVs as liquid biopsy is well-demonstrated. Further advances in the research of EV biology, characterization, and analysis could be instrumental in promoting their clinical application in the diagnosis and prognosis of cancer. Additional studies and efforts are warranted to promote the research of EVs as noninvasive biomarkers for different clinical applications. 

## Figures and Tables

**Figure 1 cancers-16-02462-f001:**
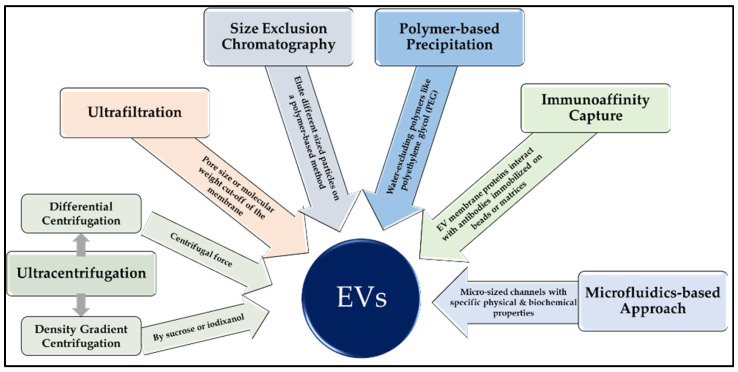
A representative chart showing different methods for isolation of extracellular vesicles (EVs) from body fluids.

**Table 4 cancers-16-02462-t004:** Extracellular Vesicles (EV) are associated with RNA and DNA as tumor biomarkers.

Cancer Type	Biomarker(s)	RNA/DNA	Isolated From	Remarks	Reference
Glioblastoma	EGFRvIII (mutant EGFR)	RNA	Serum	Detected in 28% of the patients Not detectable in patients who had undergone surgical removal of the tumor	[78]
Ovarian Cancer	Matrix Metallopeptidase 1 (MMP1)		Ascites	Patients suffering from aggressive phenotype with poor prognosis	[79]
Pan-cancer Biomarker	Human telomerase reverse transcriptase (hTERT)		Serum	67% of cancer patients compared to healthy controls High levels are associated with disease progression	[81]
Prostate Cancer	Androgen-receptor splice variant 7 (AR-V7)		Urine	Higher AR-V7 and lower AR-FL expressions in CRPC patients	[80]
Colorectal Cancer	Transforming Growth Factor Beta Receptor Type 2 (TGFBR2)	DNA	Cell lines	Frameshift mutations of TGFBR2 were detected in CRC-derived exosomes	[82]
Pancreatic Cancer	KRAS and p53 genes		Cell lines Serum	Determine genomic DNA mutations in pancreatic cancer	[83]
	KRAS gene		Plasma	Mutation frequency > 1% is associated with decreased survival probability of disease-free patients post-treatment	[84]
	NOTCH1 and BRCA2 genes		Peripheral blood and pleural effusion	Gene mutation was identified in exosomes from cancer patient	[85]
Prostate Cancer	MLH1, PTEN, and TP53 genes		Plasma	EVs showed the presence of genomic DNA fragments in different sub-types of EVs, including microvesicles and apoptotic bodies	[86]

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
