# Peer review of "Tumor-Derived Extracellular Vesicles as Liquid Biopsy for Diagnosis and Prognosis of Solid Tumors: Their Clinical Utility and Reliability as Tumor Biomarkers"

_cancers, 2024, doi:10.3390/cancers16132462_

Round 1

Reviewer 1 Report

Comments and Suggestions for Authors

The MS contains 4 very informative and interesting tables which is usefull for everyone who works with exosomes. The data are also reviewed very well noting advantages and disadvantages of the methods and possible applications. The plan of the MS is well done and easily readable. The MS mentioned ectosomes but nothing was described abount them, could the authors comment a bit the ectosomes? 

Author Response

Reviewer 1

Comments and Suggestions for Authors

The MS contains 4 very informative and interesting tables which is usefull for everyone who works with exosomes. The data are also reviewed very well noting advantages and disadvantages of the methods and possible applications. The plan of the MS is well done and easily readable. The MS mentioned ectosomes but nothing was described abount them, could the authors comment a bit the ectosomes?

Response: We highly appreciate the very positive comments by the respected reviewer. We included a very important statement about the difference between ectosomes and exosomes with two references (see Page 2)

“The origin of exosomes and ectosomes are different and although their contents have some similarities, each type has its unique membrane and cargo contents (reviewed in (Meldolesi 2018, Meldolesi 2022))”.   

Reviewer 2 Report

Comments and Suggestions for Authors

The authors wrote one comprehensive review in extracellular vesicles (EVs) based biopsy for cancer disease. They discussed different aspects of the scope, including EV preparation, using EVs as liquid biopsy (protein, miRNA, mRNA, DNA, and lipids), and the machine learning part. This is already an all-in-one review, and some minor modifications may further improve the overall readability of the article.

1. ctDNA detection is still the major approach for liquid biopsy. The author may provide some description to compare the EVs based biopsy with routine ctDNA detection. Since EVs based biopsy may pave the way for future utilization in determining cancer progression, it is better to explain why we need to develop a novel method.

2. An EVs-based assay for RNA detection might be interesting. Describing whether the procedures may affect RNA integrity is essential, as RNA is not as stable as DNA or protein. Would the authors have some information included for this?

3. The authors described different research objects using EVs based assay. Which object might be the mainstream for future development? Are there any differences between these detections with protein, mRNA, miRNA, DNA, or lipids? In the discussion and conclusion part, the authors may describe their insights in this.

4, The authors should reformat table 1 as many contents are not aligned properly.

Author Response

Reviewer 2

Comments and Suggestions for Authors

The authors wrote one comprehensive review in extracellular vesicles (EVs) based biopsy for cancer disease. They discussed different aspects of the scope, including EV preparation, using EVs as liquid biopsy (protein, miRNA, mRNA, DNA, and lipids), and the machine learning part. This is already an all-in-one review, and some minor modifications may further improve the overall readability of the article.

  1. ctDNA detection is still the major approach for liquid biopsy. The author may provide some description to compare the EVs based biopsy with routine ctDNA detection. Since EVs based biopsy may pave the way for future utilization in determining cancer progression, it is better to explain why we need to develop a novel method.

Response: We appreciate the reviewer comment about discussing the advantages of using sEVs over ctDNA in the review. We included this part as requested (3. Use of EV as liquid biopsy; page# 5).

“sEVs and circulating tumor DNA (ctDNA) are both essential components of liquid biopsies used in cancer diagnostics and prognostications (Dang and Park 2022, Yu, Li et al. 2022). However, sEVs offer several advantages over ctDNA. The sEV cargo contain RNA that increases the total number of mutant copies available for sampling compared to ctDNA alone (Yu, Hurley et al. 2021). The homogeneous size of these vesicles makes their detection easy by electron microscopy (Marrugo-Ramirez, Mir et al. 2018). The cargo contents of lipid bilayer sEVs are more stable, making them more robust for analysis compared to ctDNA (Wang, Wang et al. 2023). Moreover, the possibility of identifying gene mutations in sEVs is high compared to ctDNA (Wen, Pu et al. 2022). Additionally, sEV-associated mRNA actively released from donor cells compared to ctDNA released by necrotic or apoptotic cells (Bang, Shim et al. 2022).”

  1. An EVs-based assay for RNA detection might be interesting. Describing whether the procedures may affect RNA integrity is essential, as RNA is not as stable as DNA or protein. Would the authors have some information included for this?

Response: We totally agreed with the reviewer’s suggestion. We included a new part in page 12 as shown below.

“The method of RNA isolation may affect the sEVs yield, purity and stability, especially RNA content. So, it is very important to select the method of RNA isolation according to the study design and availability of body fluids. For example, a pure column method produces high RNA yield compared to phenol extraction method (Eldh, Lotvall et al. 2012). Another research group reported that EVs-associated RNA yield is high when isolated by membrane affinity column versus conventional ultracentrifugation method (Gutierrez Garcia, Galicia Garcia et al. 2020).”

  1. The authors described different research objects using EVs based assay. Which object might be the mainstream for future development? Are there any differences between these detections with protein, mRNA, miRNA, DNA, or lipids? In the discussion and conclusion part, the authors may describe their insights in this.

 Response: We highly appreciate the constructive comment by the reviewer. We discussed this part in page 20 (section 6: Conclusions and future prospective) and as shown below:

“To determine the most effective detection method for exosome proteins, mRNA, miRNA, DNA, and lipids, various studies have been conducted focusing on different aspects of sEVs analysis. For example, the detection of sEVs-associated miRNAs has been a prominent area of research. Several studies have highlighted the advantages of utilizing sEVs for miRNA detection. In addition to the traditional methods for miRNA detection, innovative techniques have been developed to enhance the detection sensitivity and efficiency of sEV-associated miRNAs (Lin, Jiang et al. 2022). These methods offer advantages such as cost-effectiveness, non-invasiveness, and high sensitivity, making them promising tools for biomarker development. The developed methods are continuously evolving to establish more efficient and reliable techniques for EV cargo detection.”

4, The authors should reformat table 1 as many contents are not aligned properly.

Response: We apologize for the formatting issue of Table 1. We have tried to align the table appropriately according to the style of “Cancers” and we will be sure it is in a presentable format before publication.

Reviewer 3 Report

Comments and Suggestions for Authors

This paper introduces 'Tumor-derived extracellular vesicles as liquid biopsies for the diagnosis and prognosis of solid tumors: clinical utility and reliability as tumor biomarkers'. In this study, the authors discuss the current approaches of extracellular vesicle (EV) isolation and the latest technologies of using EV-associated proteins, miRNAs, mRNAs, DNA and lipids as liquid biopsies and their advantages for clinical applications, as well as recent advances in machine learning as a new tool for tumor marker discovery. Therefore, it is suitable for publication in the journal 'MDPI-cancers' to discuss an interesting topic, but with the following modifications. The revisions are as follows, which must be confirmed before publication.

Major revisions

1. Adding a figure in the text that explains how extracellular vesicles (EVs) are isolated or the mechanism of the biomarker they contain would make it easier for the reader to understand.

Minor revisions

1. In page 2, line 88, put a period at the end of the sentence.

Author Response

Reviewer 3

Comments:

This paper introduces 'Tumor-derived extracellular vesicles as liquid biopsies for the diagnosis and prognosis of solid tumors: clinical utility and reliability as tumor biomarkers'. In this study, the authors discuss the current approaches of extracellular vesicle (EV) isolation and the latest technologies of using EV-associated proteins, miRNAs, mRNAs, DNA and lipids as liquid biopsies and their advantages for clinical applications, as well as recent advances in machine learning as a new tool for tumor marker discovery. Therefore, it is suitable for publication in the journal 'MDPI-cancers' to discuss an interesting topic, but with the following modifications. The revisions are as follows, which must be confirmed before publication.

Major revisions

  1. Adding a figure in the text that explains how extracellular vesicles (EVs) are isolated or the mechanism of the biomarker they contain would make it easier for the reader to understand.

Response: We thank the respected reviewer for the positive comments, and we completely agree with his excellent comment. Therefore, we added a new figure “figure 1” which represents the different methods used for EVs isolation (see Page 5). 

Minor revisions

  1. In page 2, line 88, put a period at the end of the sentence.

Response: We apologize for this error. We fixed the sentence and reviewed the whole manuscript to be sure it reads better.

Reviewer 4 Report

Comments and Suggestions for Authors

Comments to the author

1.     Check the grammatical mistakes in whole manuscript, for e.g., in introduction section, from line 32-38, added a colon after "subtypes", use semicolon accordingly, and Changed "programmed cells death" to "programmed cell death". Also check line 88, 139,... Do not stick to these errors only, check the entire manuscript.

2.     Author used the abbreviation for 'small extracellular vesicles' (sEVs or exosomes) later, although you mentioned 'small extracellular vesicles' earlier in the introduction. Please introduce the abbreviation first before using it. Reference 3 is very old. I would suggest authors to cite recent and update aricle…. https://www.mdpi.com/2227-9059/9/10/1373 .

3.     Some sentences are long and complex, which might affect readability. Consider breaking down longer sentences for better clarity.

4.     The author effectively lists numerous protein biomarkers identified in EVs from different cancer types in line 120-209. However, it would be beneficial to include each cancer type (e.g., Pancreatic Cancer, Colorectal Cancer, etc.) through subheadings. This would improve readability and allow readers to quickly find relevant information.

5.     The information about “miRNAs as biomarkers” from Lines 211-273, is extensive and informative. However, it would benefit from summarizing key points at the end, highlighting the most promising miRNA biomarkers for specific cancers, their diagnostic/prognostic value, and current limitations or areas needing further research.

I would suggest authors to please check this recent article https://www.mdpi.com/2072-6694/16/12/2179 and cite if possible.  

6.     Verify if the references in Table 1 are in the correct positions.

7.     Add reference in line 124-126.

8.     Please include a separate section for Limitations and challenges for EVs as therapeutics in medical field.   

Comments on the Quality of English Language

Minor English editing is required. 

Author Response

Reviewer 4

Comments to the author

  1. Check the grammatical mistakes in whole manuscript, for e.g., in introduction section, from line 32-38, added a colon after "subtypes", use semicolon accordingly, and Changed "programmed cells death" to "programmed cell death". Also check line 88, 139,... Do not stick to these errors only, check the entire manuscript.

Response: We thank the respected reviewer for his invaluable comments. We apologize for typos and sentence structure. We made the requested editing as noted.

  1. Author used the abbreviation for 'small extracellular vesicles' (sEVs or exosomes) later, although you mentioned 'small extracellular vesicles' earlier in the introduction. Please introduce the abbreviation first before using it. Reference 3 is very old. I would suggest authors to cite recent and update aricle…. https://www.mdpi.com/2227-9059/9/10/1373 .

Response: We appreciate the reviewer comment about the updated reference. We made the requested changes, and we replaced the old reference with the new one (Karn et al. 2021) as requested.

  1. Some sentences are long and complex, which might affect readability. Consider breaking down longer sentences for better clarity.

Response: We comply with the reviewer comment, and we rephrased these sentences.

  1. The author effectively lists numerous protein biomarkers identified in EVs from different cancer types in line 120-209. However, it would be beneficial to include each cancer type (e.g., Pancreatic Cancer, Colorectal Cancer, etc.) through subheadings. This would improve readability and allow readers to quickly find relevant information.

Response: We totally agree with the reviewer suggestion, and we included new tables (tables 2 to 4) alphabetically categorized based on each cancer type.

  1. The information about “miRNAs as biomarkers” from Lines 211-273, is extensive and informative. However, it would benefit from summarizing key points at the end, highlighting the most promising miRNA biomarkers for specific cancers, their diagnostic/prognostic value, and current limitations or areas needing further research.

I would suggest authors to please check this recent article https://www.mdpi.com/2072-6694/16/12/2179 and cite if possible.

Response: We thank the reviewer for this suggestion. We included this reference in the references list (Ref#53, page 12). We also added a new section of current limitations of using sEVs in clinical applications (see page 19; section 5).

  1. Verify if the references in Table 1 are in the correct positions.

Response: We thank the reviewer for this critical note. We reformatted the table to read better. We apologize of the raised issue.

  1. Add reference in line 124-126.

Response: We added a new reference 8 (Meldolesi J., 2022) in this section.

  1. Please include a separate section for Limitations and challenges for EVs as therapeutics in medical field.

Response: We took the Reviewer’s comment very seriously and we added a new section 5 (see page 19).